# Gender and Age Differences in Anthropometric Characteristics of Taiwanese Older Adults Aged 65 Years and Older

**DOI:** 10.3390/healthcare11091237

**Published:** 2023-04-26

**Authors:** Yan-Jhu Su, Chien-Chang Ho, Po-Fu Lee, Chi-Fang Lin, Yi-Chuan Hung, Pin-Chun Chen, Chang-Tsen Hung, Yun-Chi Chang

**Affiliations:** 1Department of Gerontology, University of Massachusetts Boston, Boston, MA 02125, USA; 2Department of Physical Education, Fu Jen Catholic University, New Taipei City 24205, Taiwan; 3Research and Development Center for Physical Education, Health and Information Technology, College of Education, Fu Jen Catholic University, New Taipei City 24205, Taiwan; 4Sports Medicine Center, Fu Jen Catholic Hospital, New Taipei City 243, Taiwan; 5Department of Leisure Industry and Health Promotion, National Ilan University, Yilan County 260, Taiwan; 6College of Humanities and Management, National Ilan University, Yilan County 260, Taiwan; 7Department of Physical Education and Sport Sciences, National Taiwan Normal University, Taipei City 106, Taiwan; 8Department of Sport Management, National Taiwan University of Sport, Taichung City 404, Taiwan; 9Sports Administration, Ministry of Education, Taipei City 104, Taiwan; 10Graduate Institute of Sports Training, University of Taipei, Taipei City 111, Taiwan; 11Department of Health and Leisure Management, Yuanpei University of Medical Technology, Hsinchu City 306, Taiwan

**Keywords:** anthropometry, body composition, obesity, elderly, Taiwan

## Abstract

A previous study provided reference data on the age and gender distribution of anthropometric parameters in Taiwanese adults. However, there are very few large-scale analyses of anthropometric data of older adults in Taiwan. Therefore, the main purpose of this study was to describe gender- and age-specific distributions of anthropometric measurements and anthropometry assessments of Taiwanese older adults. This was a cross-sectional study conducted on 22,389 adults aged 65 years or older (8017 males and 14,372 females) who participated in Taiwan’s National Physical Fitness Survey 2014–2015. All participants were recruited using stratified convenience sampling from 46 physical fitness test stations in 22 cities or counties in Taiwan. The anthropometric measurements and anthropometry assessments included in the present study were the assessments of height, weight, body mass index (BMI), waist circumference (WC), hip circumference (HC), and waist-to-hip ratio (WHR). The results showed statistically significant differences in weight, height, WC, and WHR across all age groups among both male and female individuals aged 65 years and older in Taiwan. However, there was no significant difference in BMI and HC between males and females in all age groups. Anthropometric status provides an initial assessment of the overall health of the population. This study uses a representative population of Taiwanese older adults over the age of 65 for analysis and provides detailed information on anthropometric data distribution.

## 1. Introduction

Taiwan’s population is aging rapidly. According to the National Development Council of the Republic of China (2020), it was estimated that the population over the age of 85 would account for 10.5% of the older adult population in 2021, and the ratio will increase to 27.4% in 2070. The World Health Organization (WHO) and the United Nations define an “aging society” as one in which the population over 65 years old accounts for 7% of the total population; up to 14% is called an “aged society”; and up to 20% is called a “superaged society”. Taiwan became an aging society in 1993 (1.49 million, 7.1%) and turned into an aged society in 2018 (3.43 million, 14.6%). It is estimated that it will become a superaged society in 2025 (4.7 million, 20.1%) [1]. Compared to other countries, Taiwan’s population is aging very rapidly [2]. Moreover, since January 2018, the demographic structure in Taiwan has begun to experience negative population growth, which means that the birth rate is lower than the mortality rate [1].

Rapid aging is accompanied by social and financial challenges in most developed countries [3]. Even countries that have established long-term care financing mechanisms still require their citizens to pay substantial costs out of their own pockets [4,5]. The cost of long-term care creates catastrophic financial risk for most families. In addition, rapid growth in government fiscal spending on the social welfare system and older adults’ health care casts doubt on the sustainability of national long-term care systems [6]. Therefore, developing a long-term care insurance plan appears to be a more practical alternative for a rapidly aging society. In developing long-term care insurance plans, the assessment of the physical condition of older adults is very important, as it affects the development of disease [7] and the risk of disease and death [8,9,10].

An individual’s physical fitness [11] and general health [8] can be evaluated through anthropometric parameters. Body measures have also been shown to be significantly correlated with an individual’s health status [9]. In addition, anthropometric measurements are an essential component of a complete assessment of older adults [12], with the advantages of being portable, noninvasive, low-cost, and easy to use [13,14].

The WHO Expert Committee recommended that older adults regularly monitor their health using anthropometric methods to know instantly if their health status has changed [15]. Furthermore, they also suggested that due to the great variability in the older adult population, the age group of at least 10 years of age should have validation data displayed by gender. Ideally, each anthropometric index and age group should also include data from the overall group [13,16,17]. A previous study provided reference data on the age and gender distribution of anthropometric parameters in Taiwanese adults [18]. However, there are few large-scale analyses of anthropometric data of older adults in Taiwan [18,19]. Therefore, the main purpose of this study was to describe gender- and age-specific distributions of anthropometric measurements and anthropometry assessments of Taiwanese older adults.

## 2. Materials and Methods

### 2.1. Study Participants, Design and Procedure

This cross-sectional study used data from Taiwan’s National Physical Fitness Survey (TNPFS) [20,21,22]. The aim of this study was to examine the health-related physical fitness status of Taiwanese individuals from October 2014 to March 2015. This study was conducted by the Sports Administration, Ministry of Education in Taiwan. The details of the design, sampling protocols, and data validation of the series of annual surveys have been described previously [13]. Briefly, all participants were recruited using stratified convenience sampling from 46 physical fitness test stations in 22 cities or counties in Taiwan. The study protocols were as follows: participants completed a standardized, structured questionnaire, followed by a pretest health screening, anthropometric measurements, and health-related physical fitness tests. The questionnaire included questions on demographic characteristics, socioeconomic status, lifestyle behaviors, current health status, and associated factors related to happiness. A pretest health screening included measurements of resting heart rate and blood pressure (systolic and diastolic), anthropometric assessments, health-related physical fitness tests (screening for health limitations affecting eligibility for participation in health-related physical fitness tests), and a modified Physical Activity Readiness Questionnaire [15]. Finally, a representative sample of Taiwanese older adults 65 years and older was included in the survey (*n* = 22,389). These data contain deidentified secondary data that have been released to the public for research purposes [17,18,19]. This study was conducted in accordance with the Declaration of Helsinki, and the protocol was approved by the Institutional Review Board of Fu Jen Catholic University in Taiwan (FJU-IRB C110113).

### 2.2. Data Collection Procedures

The data were collected by a standardized, structured questionnaire that included sociodemographic variables, self-reported health status, and lifestyle behaviors. All participants completed the questionnaire via face-to-face interviews; questions included those regarding gender (male, female); age (65–69, 70–74, 75–79, 80–84, ≥85 years); education (elementary school or lower, junior or senior high school, college or higher); monthly income (≤20,000 New Taiwan dollar [NTD], 20,001–40,000 NTD, ≥40,001 NTD); marital status (never married, married, divorced/separated/widowed); self-reported health status (excellent or good, fair, very bad or poor); smoking (never, current, former); chewing betel nut (never, current, former).

### 2.3. Anthropometric Variables Measurements

The body weight, height, waist circumference (WC), and hip circumference (HC) of each survey participant were measured as anthropometric variables; the body mass index (BMI, in kilograms per square meter) and waist-to-hip ratio (WHR) were calculated after measuring body weight, height, WC, and HC. The examiners, who had attended a regional training seminar and passed a certification test, of which detailed standardized procedures have been reported in previous studies, conducted these anthropometric measurements [20,21,22]. To measure the anthropometric variables, the participants had to remove their shoes and heavy clothes. Body weight and height were recorded in kilograms and meters to the nearest 0.1 kg and 0.1 cm using an electronic height–weight scale, which has passed official inspection in Taiwan. Participants were in a standing position and exhaled while the examiners measured WC, which was measured to the nearest 0.1 cm with a flexible steel tape measure placed midway between the lowest rib and the iliac crest. HC was measured to the nearest 0.1 cm at the widest part of the hip region in the standing position.

### 2.4. Anthropometry Assessment

BMI and WHR were calculated as anthropometry assessments in this study. BMI and WHR were calculated with the following formulas–BMI = body weight kg/height squared m^2^; WHR = WC cm/HC cm–after measuring anthropometric variables, including body weight, height, WC, and HC. The Health Promotion Administration, Ministry of Health and Welfare in Taiwan suggested the cutoff BMI points for general obesity, which classified BMI into four categories: underweight (BMI < 18.5 kg/m^2^), normal weight (18.5 ≤ BMI < 24 kg/m^2^), overweight (24 ≤ BMI < 27 kg/m^2^) and obese (BMI ≥ 27 kg/m^2^) [12]. In addition, the WC points were divided into two categories to evaluate whether participants had abdominal obesity. WC was calculated using the following cutoff points: for males, a WC ≥ 90 cm, and for females, a WC ≥ 80 cm [20].

### 2.5. Statistical Analysis

This study used SAS 9.4 (SAS Institute, Cary, NC, USA) to analyze the data. The anthropometric measurements were calculated for gender- and age-group-specific (65–69, 70–74, 75–79, 80–84, and ≥85 years) means, standard deviations, and percentiles (5th, 10th, 15th, 25th, 50th, 75th, 85th, 90th, 95th). Continuous variables were analyzed using Student’s *t* test or analysis of variance (ANOVA), and the chi-square test was used to analyze categorical variables. If the F value was significant (*p* < 0.05), Tukey’s post hoc test was performed to determine the differences between the pairs of means. Statistical results were considered to be significant at *p* < 0.05.

## 3. Results

Table 1 shows the demographic and anthropometric characteristics of the study population. The final sample consisted of 22,389 adults aged 65 years and older, of which 64.19% were females (*n* = 14,372). Their ages ranged from 65 to 103 years, with a mean of 73.29 (SD = 6.43) years. For females, the mean age was 72.69 (SD = 6.11) years; for males, it was 74.36 (SD = 6.83) years. In this sample, 14.22% had completed college or higher education, 28.91% completed junior or senior high school, and more than half (56.87%) had completed elementary school or lower education; 64.29% females reported having elementary school or lower education, 26.68% reported completing junior or senior high school, and 9.03% reported having college or higher education. For males, the reported values were 43.69%, 32.86%, and 23.45%, respectively; these differences in educational attainment were statistically significant (*p* < 0.001). Moreover, 74.64% of females in the sample were single (including those who were divorced, separated, or widowed) at the time of the survey; the proportion of those who were married was 25.36%. Similarly, 69.96% of the males were single (including those who were divorced, separated, or widowed); 30.04% reported being married. These differences were statistically significant (*p* < 0.001). Regarding self-reported health status, most participants felt excellent or good (females 63.65%; males 68.27%); 27.08% of females and 24.87% of males felt fair; 8.55% of females and 6.86% of males felt very bad or poor. These differences in self-reported health status were statistically significant (*p* < 0.001). Regarding smoking status, 8.42% of males and 0.81% of females used to smoke, and 11.13% of males and 1.76% of females currently smoked. In contrast, 80.45% of males and 97.44% of females never smoked; these differences in smoking status were statistically significant (*p* < 0.001). Regarding betel nut chewing, 3.29% of males and 0.37% of females used to chew betel nut, and 1.83% of males and 0.72% of females currently chewed betel nut. In contrast, 94.89% of males and 98.92% of females had never chewed betel nut; these differences in betel nut chewing status were statistically significant (*p* < 0.001). ANOVA showed a statistically significant difference by age group for body weight, height, and BMI (Table 2 and Table 3), as well as for WC, HC, and WHR (*p* < 0.05) (Table 4 and Table 5). In addition, statistically significant differences were found between males and females in body weight, height, BMI, WC, and WHR (*p* < 0.05). However, there was no significant difference in HC between males and females (*p* = 0.747).

Table 2 and Table 3 show the results for weight, height, and BMI (mean, standard deviation, CV, and percentiles) for males and females, respectively, distributed by age group. For both females and males, mean weight and height were lower in the oldest age groups when compared with the youngest (*p* < 0.05), and the weight in females was lower. A multiple comparison procedure using Tukey’s test showed that for males, differences were significant among all groups, with the exception of the 70- to 74-year-old and 75- to 79-year-old groups (Table 2); for females, differences were significant among all groups for body weight (Table 3). For height, the differences were significant among all groups in males, except for the 75–79 and 80–84 age groups (Table 2); for females, the differences were significant among all groups for body weight (Table 3).

Mean BMI was lower with increasing age (*p* < 0.05) in males. Multiple comparisons showed that both males and females showed no significant difference between the three groups (65–69, 70–74, 75–79 age groups) (Table 2 and Table 3). Females had significant differences in all the other groups (80–84, ≥ 85 age groups) for BMI (Table 3). However, there was no significant difference in males between the 80–84 and the ≥ 85 age groups (Table 2).

Distributions of body circumference (means, standard deviations, percentiles, and CV) are shown in Table 4 and Table 5. The mean WC values for males were higher than those for females in all age groups. For males, the results showed a significant difference between two groups (the 65–69 and 75–79 age groups), but, as with the HC results, there were significant differences between the 65–69 and 75–79 age groups. For females, there were no significant differences between the three groups (75–79, 80–84, and ≥85 years) but there were significant differences in the other groups (65–69 and 70–74 years) for WC. On the other hand, HC was significantly different in the ≥85 years age group when compared to all the other groups (65–69, 70–74, 75–79, 80–84 age groups). The mean WHR values for males were higher than those for females. For males, the 70–74 and 75–79 age groups were significantly different to the 65–69, 80–84, and ≥85 age groups in terms of HC. In contrast, HC in females was significantly different in all the age groups.

Table 6 presents the prevalence of BMI categories according to the World Health Organization cutoff points [12]. The results showed that both males and females with normal weight had the highest prevalence (males 39.18%, females 40.89%). However, the ratio decreased with increasing age in normal weight, overweight, and obese individuals. Table 7 presents the prevalence of WC categories according to the World Health Organization cutoff points (males 90 cm and females 80 cm) [23]. The results show that males have a higher ratio of the WC < 90 (cm) cutoff point (52.53%), but females have a higher ratio of the WC ≥ 80 (cm) cutoff point (69.10%). Table 8 shows the prevalence of WHR categories according to the suggested cutoff point (males 0.9 and females 0.85) of the Health Promotion Administration in Taiwan. The results show that both males and females have a higher prevalence than the suggested cutoff point (males 64.72%, females 64.17%).

## 4. Discussion

The aim of this study was to summarize anthropometric measurements in a representative sample of older Taiwanese adults aged 65 years and older. The results showed statistically significant differences in weight, height, WC, and WHR across all age groups among both male and female individuals aged 65 years and older in Taiwan. However, there was no significant difference in BMI and HC between males and females in all age groups. In addition, males had higher mean WC values than females in all age groups, but the difference was statistically significant in two age groups (65–69 and 75–79 years). The HC results were similar in the same two age groups. The average WHR of males in all age groups was higher than that of females. The 75- to 79-year-old group was significantly different from the 65- to 69- and ≥85-year-old groups in the male group. In addition, there was also a statistically significant difference between the 65–69 and 70–74 age groups in the male group. In contrast, there were significant differences among females across all age groups.

According to the World Health Organization’s BMI classification criteria, a BMI greater than 24 is considered overweight, and a BMI greater than 27 is considered obese [13]. Although the results show that both males and females with normal weight had the highest prevalence (males 39.87%, females 41.66%), 57.88% of males and 56.17% of females were in the overweight and obese categories. This means that overweight and obesity are obvious problems in Taiwan’s older adult population. A Mexican study of anthropometric measurements of the older population noted that female BMI measurements had a higher prevalence of obesity among older Mexicans and a significant difference from males [24]. Similar results were also illustrated in the United States [25] and other countries [26]. However, our results show that the percentages of males and females in these two categories are not very different. This may also be caused by cultural [27] and lifestyle differences [28] between Eastern and Western countries.

Although BMI is widely used as an indicator of obesity [29], some individuals with a BMI in the normal category may still be in the obese category for other body measurements [30]. As a result, body shape or anthropometry assessments may be a better indicator of obesity [31]. Individuals with a higher WHR may have a higher risk of abdominal obesity [32]. Abdominal obesity is significantly associated with many chronic diseases [33]. In a previous study on Taiwanese adults, WHR increased with age [16]. Moreover, according to the recommendation of the Health Promotion Administration of the Republic of China [34], males with a WC greater than or equal to 90 cm and females with a WC greater than 80 cm are considered to have abdominal obesity. According to our data, 48.02% of males and 30.10% of females had a WC of ≥90/80. Moreover, males with a WHR greater than or equal to 0.9 and females with a WHR greater than 0.85 cm are considered to have abdominal obesity. According to our data, 64.72% of males and 64.17% of females had a WHR of ≥0.9/0.85. However, in the classification of BMI, 21.74% of males and 23.91% of females classified as obese were over 65 years old. There are significant differences in such ratios. Overall, our study provides WC, HC, and WHR data as a reference. Previous studies show that abdominal obesity is associated with an increased risk of several chronic diseases, including cardiovascular disease [35], diabetes [36], asthma [37], and cancer [38]; this is especially the case in older populations [35,36]. As a result, these data can reveal possible health risks to Taiwanese older adults. Future research could focus on specific anthropometric characteristics and their associations with the risk of different chronic diseases. In addition, helping the population adhere to a healthier lifestyle is also essential for national development [16]. We believe that cross-referencing several indicators can lead to more accurate population characteristics.

We noticed that 65% of older adults in our sample had a WHR greater than the cutoff point. We believe this may be the result of loss of muscle and reduced activity in the older population. Loss of muscle can lead to changes in body shape. In future studies, we suggest that anthropometry assessment analysis be added to further understand anthropometry assessment ratios, which refer to the percentage of fat, bone, and muscle in the body of older adults in Taiwan, and to compare whether the anthropometry assessment ratios of older adults are different from those of adults. In addition, the anthropometric characteristics described above can also be analyzed alongside disease to establish better criteria and cutoff points for the Taiwanese population.

Despite the above outcomes, there were still some limitations in this study. First, although this study collected age and gender data, it used a cross-sectional design, so it was difficult to establish a causal relationship between gender- and age-specific anthropometrics. Therefore, it is recommended that future studies adopt an experimental design to confirm the causal relationship. Second, this study recruited people over 65 years old, and as such cannot effectively estimate the anthropometric measurements of the Taiwanese population. For that reason, future studies can comprehensively present population groups of different ages, such as young adults, children, and infants, to establish a wider range of anthropometric data; this will be necessary to increase the amount of information on the entire Taiwanese population in the future. Third, anthropometric measurements include many different factors; to have complete information on the entire population, it will be necessary to collect more anthropometric data. Last, these data do not provide older individuals’ medical history or current medical condition. Older adults are a relatively vulnerable group, and the rate of disease among older adults is much higher than that among adults [3]. Studies have also demonstrated that whether older adults have a disease affects an individual’s physical activity and anthropometric status [35]. It is recommended to add an individual’s disease history and current disease status to the database to facilitate future analysis and research.

In addition to the above suggestions, future studies could use a longitudinal study design to examine causality after receiving more waves of data. Assessments of lifestyle and physical activity in older adults can be included as well. Anthropometric measurements of older adults are greatly influenced by their lifestyle [12]. Studies have pointed out that after several weeks of Pilates exercise, older females have significant improvements in BMI, waist-to-hip ratio, body fat percentage, visceral fat, and other related obesity measurement [39]. In addition, a study has also shown that aquatic exercise programs have a positive impact on the health of older people with unwell physical conditions [40]. Therefore, physical activity may have influenced the anthropometric results. In future research design and data collection, it is recommended to include the frequency and type of physical activity of the older adults for examination.

In summary, this study recruited anthropometric and anthropometry assessment data of adults over 65 years of age to evaluate gender- and age-specific distributions. However, several studies have investigated the relationship between anthropometric measurements and products/workplaces [41,42,43]. Measured variables, including BMI, waist circumference, hip circumference, and waist-to-hip ratio, will be decisive in predicting the most common problems in older adults, such as fatty liver, high blood pressure, etc. [44]. Based on the current study, anthropometric measurements of older adults can be used to bridge the gap at the first stage of assistive device or product design to improve comfort and practicality. Furthermore, the development of assistive technology involves the expansion of functional abilities and the restoration of deficient functions or help in carrying out activities [45]; these are important to improve the quality of life and well-being of older people who live alone. In a rapidly aging population, this will be an essential issue for anthropometric measurement collection and related industry development in Taiwan.

## 5. Conclusions

In conclusion, anthropometric status provides an initial assessment of the overall health of the population. This study uses a representative population of Taiwanese older adults over the age of 65 for analysis and provides detailed information on anthropometric data distribution. The results of these data reveal the profile of Taiwan’s older adult population and provide a demographic basis for future research. Additionally, these data may provide authorities with an adequate overview for clinical and theoretical purposes.

## Figures and Tables

**Table 1 healthcare-11-01237-t001:** Demographic and anthropometric characterization of study population.

Variables	Total (*N* = 22389)	Males (*n* = 8017)	Females (*n* = 14372)	*p*-Value
Age (years)	73.29 ± 6.43	74.36 ± 6.83	72.69 ± 6.11	<0.0001 *
Education level (%)				<0.0001 *
Elementary school or lower	56.87	43.69	64.29	
Junior or senior school	28.91	32.86	26.68	
College or higher	14.22	23.45	9.03	
Income level (%)				<0.0001 *
≦20,000 NTD	86.27	78.96	90.37	
20,001–40,000 NTD	7.99	11.32	6.12	
≧40,001 NTD	5.74	9.72	3.51	
Marital status (%)				<0.0001 *
Never married	53.53	58.78	49.02	
Married	27.04	30.04	25.36	
Divorced/separation/widowed	20.43	11.18	25.62	
Self-reported health status (%)				<0.0001 *
Excellent or good	65.31	68.27	63.65	
Fair	26.75	24.87	27.08	
Very bad or poor	7.94	6.86	8.55	
Smoking status (%)				<0.0001 *
Never	91.34	80.45	97.44	
Current	5.12	11.13	1.76	
Former	3.54	8.42	0.81	
Chewing betel nut (%)				<0.0001 *
Never	97.47	94.89	98.92	
Current	1.11	1.83	0.72	
Former	1.41	3.29	0.37	

Abbreviations: NTD, New Taiwan Dollar; SD, standard deviation. Values are expressed as means ± SD. * Significantly different between males and females by student’s *t*-test or chi-square test at *p* < 0.05.

**Table 2 healthcare-11-01237-t002:** Body weight, height, and BMI in males aged 65 years and older (Numbers; mean values and SD).

Variables	*n*	Mean	SD	%CV	p5	p10	p15	p25	p50	p75	p85	p90	p95
Body weight (kg) †													
65–69 *	2333	67.2 ^a^	8.8	13.0	53.0	56.0	58.0	61.0	67.0	73.0	77.0	79.0	82.0
70–74 *	1978	66.0 ^b^	8.6	13.0	52.0	55.0	57.0	60.0	66.0	72.0	75.0	77.0	80.0
75–79 *	1698	65.5 ^b^	9.0	13.7	50.3	54.0	56.0	59.0	65.5	72.0	75.0	77.0	80.0
80–84 *	1106	64.2 ^c^	9.0	13.9	50.0	52.9	55.0	58.0	64.0	70.0	73.2	76.0	79.1
≧85 *	764	62.4 ^d^	9.0	14.4	47.3	51.0	53.0	56.0	63.0	69.0	72.0	74.0	77.0
Total *	7879	65.7	8.9	13.6	51.0	54.0	56.0	59.5	66.0	72.0	75.0	77.6	80.0
Height (cm) †													
65–69 *	2368	164.3 ^a^	5.9	3.6	154.0	157.0	158.0	161.0	164.7	168.3	170.1	172.0	174.0
70–74 *	2007	163.6 ^b^	6.0	3.7	153.0	156.0	157.5	160.0	164.0	168.0	170.0	171.0	173.0
75–79 *	1725	163.0 ^c^	6.0	3.7	153.0	156.0	157.0	159.0	163.0	167.0	169.0	171.0	173.0
80–84 *	1107	162.7 ^c^	5.9	3.6	153.0	155.0	157.0	159.0	163.0	167.0	169.0	170.0	172.1
≧ 85 *	770	161.7 ^d^	6.2	3.8	152.0	154.0	156.0	158.0	162.0	166.0	168.0	170.0	172.0
Total *	7977	163.4	6.0	3.7	153.0	156.0	157.0	160.0	164.0	167.2	170.0	171.0	173.0
BMI (kg/m^2^) †													
65–69	2314	24.9 ^a^	3.0	12.0	20.1	21.1	21.8	22.9	24.8	26.9	28.0	28.8	30.0
70–74 *	1970	24.7 ^a^	2.9	11.9	19.9	20.9	21.7	22.7	24.6	26.7	27.8	28.6	29.6
75–79 *	1690	24.7 ^a^	3.1	12.5	19.5	20.7	21.5	22.7	24.5	26.8	27.9	28.9	30.0
80–84	1102	24.2 ^b^	3.0	12.5	19.0	20.4	21.1	22.3	24.2	26.2	27.3	28.1	29.4
≧85	764	23.9 ^b^	3.1	12.8	18.7	19.7	20.4	21.8	24.0	26.0	27.2	27.8	28.7
Total *	7840	24.6	3.0	12.3	19.6	20.7	21.5	22.6	24.5	26.6	27.7	28.6	29.7

Abbreviations: BMI, body mass index; CV, coefficient of variance; SD, standard deviation. p5, p10, p15, p25, p50, p75, p85, p90 and p95, 5th, 10th, 15th, 25th, 50th, 75th, 85th, 90th and 95th percentiles. ^a,b,c,d^ For each measure, superscript letters next to the mean values indicate results of Tukey’s test. Means sharing a letter indicate that values between the age groups are not significantly different from each other, whereas mean values with different superscript letters indicate that they are significantly different from each other. Mean values that show two superscript letters indicate that the specific age group is significantly different from one group(s) but not significantly different from the other(s) (*p* < 0.05). * Mean values were significantly different between males and females (Student’s *t* test; *p*< 0.05). † Mean values were significantly different across all age groups (ANOVA; *p* < 0.05).

**Table 3 healthcare-11-01237-t003:** Body weight, height, and BMI in females aged 65 years and older (Numbers; mean values and SD).

Variables	*n*	Mean	SD	%CV	p5	p10	p15	p25	p50	p75	p85	p90	p95
Body weight (kg) †													
65–69 *	5297	58.8 ^a^	8.7	14.8	45.7	48.0	50.0	53.0	58.0	65.0	68.0	70.0	74.0
70–74 *	3995	58.3 ^b^	8.4	14.4	45.0	48.0	50.0	52.3	58.0	64.0	67.0	70.0	73.0
75–79 *	2894	57.3 ^c^	8.7	15.2	44.0	46.0	48.0	51.0	57.0	63.0	67.0	69.0	72.0
80–84 *	1496	55.4 ^d^	8.4	15.2	42.0	45.0	47.0	50.0	55.0	60.6	64.0	67.0	78.0
≧85 *	647	53.3 ^e^	8.3	15.6	41.0	43.0	45.0	47.1	52.0	59.0	62.0	65.0	68.0
Total *	14,329	57.8	8.7	15.0	44.0	47.0	49.0	52.0	57.0	63.0	67.0	69.3	73.0
Height (cm) †													
65–69 *	5310	154.1 ^a^	5.6	3.6	145.0	147.0	148.0	150.0	154.0	158.0	160.0	161.0	163.0
70–74 *	4008	152.9 ^b^	5.5	3.6	144.0	146.0	147.0	149.0	153.0	156.0	158.1	160.0	162.0
75–79 *	2895	151.7 ^c^	5.7	3.8	142.0	144.0	146.0	148.0	152.0	155.0	157.0	159.0	161.0
80–84 *	1497	150.7 ^d^	5.5	3.7	142.0	144.0	145.0	147.0	151.0	154.0	156.0	157.2	160.0
≧ 85 *	649	149.5 ^e^	6.1	4.1	140.0	142.0	143.0	145.0	149.5	154.0	156.0	157.0	159.0
Total *	14,359	152.7	5.8	3.8	143.0	145.0	147.0	149.0	152.4	156.0	158.8	160.0	162.0
BMI (kg/m^2^) †													
65–69	5295	24.8 ^a^	3.5	14.1	19.6	20.5	21.3	22.4	24.5	26.9	28.5	29.4	31.1
70–74 *	3992	25.0 ^a^	3.4	13.7	19.8	20.8	21.5	22.6	24.7	27.1	28.5	29.5	31.1
75–79 *	2891	24.9 ^a^	3.6	14.3	19.3	20.5	21.2	22.4	24.7	27.2	28.5	29.6	31.2
80–84	1494	24.4 ^b^	3.5	14.4	18.9	20.1	20.8	21.9	24.2	26.6	28.2	29.1	30.5
≧85	645	23.9 ^c^	3.3	14.0	18.7	19.8	20.6	21.6	23.5	25.9	27.2	28.7	30.2
Total *	14,317	24.8	3.5	14.1	19.5	20.5	21.2	22.4	24.5	27.0	28.4	29.4	31.0

Abbreviations: BMI, body mass index; CV, coefficient of variance; SD, standard deviation. p5, p10, p15, p25, p50, p75, p85, p90 and p95, 5th, 10th, 15th, 25th, 50th, 75th, 85th, 90th and 95th percentiles. ^a,b,c,d,e^ For each measure, superscript letters next to the mean values indicate results of Tukey’s test. Means sharing a letter indicate that values between the age groups are not significantly different from each other, whereas mean values with different superscript letters indicate that they are significantly different from each other. Mean values that show two superscript letters indicate that the specific age group is significantly different from one group(s) but not significantly different from the other(s) (*p* < 0.05). * Mean values were significantly different between males and females (Student’s *t* test; *p* < 0.05). † Mean values were significantly different across all age groups (ANOVA; *p* < 0.05).

**Table 4 healthcare-11-01237-t004:** WC, HC, and WHR in males aged 65 years and older (Numbers; mean values and SD).

Variables	*n*	Mean	SD	%CV	p5	p10	p15	p25	p50	p75	p85	p90	p95
WC (cm) †													
65–69 *	2387	88.6 ^a^	8.8	9.9	74.5	77.0	79.5	83.0	88.0	94.0	98.0	100.0	104.0
70–74 *	2015	89.0 ^a^	8.8	9.9	75.0	78.0	80.0	83.0	89.0	95.0	98.0	100.0	104.0
75–79 *	1733	89.9 ^b^	9.2	10.3	74.0	78.0	80.5	84.0	90.0	96.0	99.0	101.0	105.0
80–84 *	1111	89.0 ^a^	9.1	10.2	73.0	77.0	79.5	83.0	89.0	96.0	98.0	100.0	104.0
≧85 *	771	88.7 ^a^	9.1	10.2	74.0	77.5	79.0	82.0	89.0	95.0	98.0	100.0	103.0
Total *	8017	89.0	9.0	10.1	74.0	77.5	80.0	83.0	89.0	95.0	98.0	100.0	104.0
HC (cm)													
65–69	2387	96.4	6.2	6.5	87.0	89.0	90.0	92.0	96.0	100.0	102.5	104.0	107.0
70–74 *	2015	96.1	6.2	6.5	86.0	89.0	90.0	92.0	96.0	100.0	102.0	104.0	107.0
75–79	1733	96.7	6.4	6.6	86.0	89.0	90.0	93.0	96.5	101.0	103.0	105.0	107.0
80–84	1111	96.5	6.4	6.6	87.0	88.5	90.0	92.0	96.0	101.0	103.0	104.5	108.0
≧85 *	771	96.3	6.5	6.8	86.0	88.0	90.0	92.0	96.0	100.5	103.0	104.0	107.5
Total *	8017	96.4	6.3	6.5	86.5	89.0	90.0	92.0	96.0	100.0	103.0	104.0	107.0
WHR †													
65–69 *	2387	0.92 ^a^	0.06	6.4	0.83	0.85	0.86	0.88	0.92	0.96	0.98	0.99	1.01
70–74 *	2015	0.93 ^b^	0.06	6.8	0.82	0.85	0.86	0.89	0.92	0.96	0.98	1.00	1.03
75–79 *	1733	0.93 ^b^	0.07	7.2	0.83	0.85	0.87	0.89	0.93	0.97	0.99	1.01	104
80–84 *	1111	0.92 ^a^	0.07	7.2	0.82	0.84	0.86	0.88	0.92	0.97	0.99	1.00	1.03
≧85 *	771	0.92 ^a^	0.06	7.0	0.82	0.84	0.85	0.88	0.92	0.96	0.99	1.00	1.03
Total *	8017	0.92	0.06	6.9	0.82	0.85	0.86	0.88	0.92	0.96	0.98	1.00	1.03

Abbreviations: WC, waist circumference; HC, hip circumference; WHR, waist-to-hip ratio; CV, coefficient of variance; SD, standard deviation. p5, p10, p15, p25, p50, p75, p85, p90 and p95, 5th, 10th, 15th, 25th, 50th, 75th, 85th, 90th and 95th percentiles. ^a,b^ For each measure, superscript letters next to the mean values indicate results of Tukey’s test. Means sharing a letter indicate that values between the age groups are not significantly different from each other, whereas mean values with different superscript letters indicate that they are significantly different from each other. Mean values that show two superscript letters indicate that the specific age group is significantly different from one group(s) but not significantly different from the other(s) (*p* < 0.05). * Mean values were significantly different between males and females (Student’s *t* test; *p* < 0.05). † Mean values were significantly different across all age groups (ANOVA; *p* < 0.05).

**Table 5 healthcare-11-01237-t005:** WC, HC, and WHR in females aged 65 years and older (Numbers; mean values and SD).

Variables	*n*	Mean	SD	%CV	p5	p10	p15	p25	p50	p75	p85	p90	p95
WC (cm) †													
65–69 *	5312	83.4 ^a^	9.2	11.0	69.0	72.0	74.0	77.0	83.0	89.3	93.0	95.5	100.0
70–74 *	4011	84.6 ^b^	9.5	11.2	70.0	73.0	75.0	78.0	84.0	91.0	95.0	97.0	101.0
75–79 *	2898	85.9 ^c^	9.9	11.5	70.0	73.5	76.0	79.0	86.0	92.0	96.0	98.5	102.5
80–84 *	1499	86.2 ^c^	10.0	11.6	70.5	74.0	76.0	79.0	86.0	93.0	97.0	100.0	103.0
≧85 *	652	86.6 ^c^	9.7	11.2	71.0	75.0	77.0	80.0	86.0	93.0	97.0	100.0	104.0
Total *	14,372	84.7	9.6	11.3	70.0	73.0	75.0	78.0	84.0	91.0	95.0	97.5	101.5
HC (cm) †													
65–69	5312	96.3 ^a^	7.0	7.3	86.0	88.0	89.0	91.5	96.0	101.0	104.0	105.0	109.0
70–74 *	4011	96.6 ^a^	7.2	7.4	86.0	88.0	89.5	92.0	96.0	101.0	104.0	106.0	109.5
75–79	2898	96.6 ^a^	7.3	7.6	85.0	88.0	89.0	92.0	96.0	101.0	104.0	106.5	110.0
80–84	1499	96.1 ^a^	7.4	7.7	85.0	87.0	88.5	91.0	96.0	101.0	104.0	106.0	109.0
≧85 *	652	95.0 ^b^	7.3	7.7	83.5	86.0	87.5	90.0	94.0	99.7	102.5	104.0	108.0
Total *	14,372	96.3	7.2	7.5	85.1	88.0	89.0	91.0	96.0	101.0	104.0	106.0	109.0
WHR †													
65–69 *	5312	0.87 ^a^	0.07	7.7	0.76	0.79	0.80	0.82	0.86	0.90	0.93	0.95	0.98
70–74 *	4011	0.88 ^b^	0.07	8.4	0.77	0.79	0.80	0.83	0.87	0.92	0.95	0.97	0.99
75–79 *	2898	0.89 ^c^	0.07	8.2	0.78	0.80	0.82	0.84	0.89	0.93	0.96	0.98	1.01
80–84 *	1499	0.90 ^d^	0.07	8.1	0.79	0.81	0.82	0.85	0.89	0.94	0.98	0.99	1.02
≧85 *	652	0.91 ^e^	0.08	8.4	0.80	0.82	0.84	0.86	0.91	0.96	0.99	1.02	1.05
Total *	14,372	0.88	0.07	8.3	0.77	0.79	0.81	0.83	0.87	0.92	0.95	0.97	1.00

Abbreviations: WC, waist circumference; HC, hip circumference; WHR, waist-to-hip ratio; CV, coefficient of variance; SD, standard deviation. p5, p10, p15, p25, p50, p75, p85, p90 and p95, 5th, 10th, 15th, 25th, 50th, 75th, 85th, 90th and 95th percentiles. ^a,b,c,d,e^ For each measure, superscript letters next to the mean values indicate results of Tukey’s test. Means sharing a letter indicate that values between the age groups are not significantly different from each other, whereas mean values with different superscript letters indicate that they are significantly different from each other. Mean values that show two superscript letters indicate that the specific age group is significantly different from one group(s) but not significantly different from the other(s) (*p* < 0.05). * Mean values were significantly different between males and females (Student’s *t* test; *p* < 0.05). † Mean values were significantly different across all age groups (ANOVA; *p* < 0.05).

**Table 6 healthcare-11-01237-t006:** Categories of BMI in males and females aged 65 years and older (Numbers and percentages).

Variables	Age Groups											
	65–69		70–74		75–79		80–84		≧85		Total	
	*n*	%	*n*	%	*n*	%	*n*	%	*n*	%	*N*	%
Males *												
Normal 18.5–23.9 (kg/m^2^)	865	36.24	782	38.81	678	39.12	469	42.21	347	45.01	3141	39.18
Underweight <18.5 (kg/m^2^)	111	4.65	82	4.07	85	4.90	47	4.23	40	5.19	365	4.55
Overweight 24.0–26.9 (kg/m^2^)	868	36.36	720	35.73	583	33.64	409	36.81	260	33.72	2840	35.42
Obesity ≥ 27 (kg/m^2^)	543	22.75	431	21.39	387	22.33	186	16.74	124	16.08	1671	20.84
Total	2387	100.00	2015	100.00	1733	100.00	1111	100.00	771	100.00	8017	100.00
Females *												
Normal 18.5–23.9 (kg/m^2^)	2189	41.21	1582	39.44	1115	38.74	655	43.70	335	51.38	5876	40.89
Underweight < 18.5 (kg/m^2^)	135	2.54	89	2.22	80	2.76	56	3.74	34	5.21	394	2.74
Overweight 24.0–26.9 (kg/m^2^)	1693	31.87	1308	32.61	939	32.40	455	30.35	173	26.53	4568	31.78
Obesity ≥ 27 (kg/m^2^)	1295	24.38	1032	25.73	764	26.36	333	22.21	110	16.87	3534	24.59
Total	5312	100.00	4011	100.00	2898	100.00	1499	100.00	652	100.00	14372	100.00
Pooled *												
Normal 18.5–23.9 (kg/m^2^)	3054	39.67	2364	39.23	1793	38.72	1124	43.07	682	47.93	9017	40.27
Underweight < 18.5 (kg/m^2^)	246	3.20	171	2.84	165	3.56	103	3.95	74	5.20	759	3.39
Overweight 24.0–26.9 (kg/m^2^)	2561	33.26	2028	33.65	1522	32.87	864	33.10	433	30.43	7408	33.09
Obesity ≥ 27 (kg/m^2^)	1838	23.87	1463	24.28	1151	24.85	519	19.89	234	16.44	5205	23.25
Total	7699	100.00	6026	100.00	4631	100.00	2610	100.00	1423	100.00	22389	100.00

Abbreviations: BMI, body mass index. * Prevalence of BMI categories were significantly different across all age groups (chi-square test; *p* < 0.05).

**Table 7 healthcare-11-01237-t007:** Categories of WC in males and females aged 65 years and older (Numbers and percentages).

Variables	Age groups											
	65–69		70–74		75–79		80–84		≧85		Total	
	*n*	%	*n*	%	*n*	%	*n*	%	*n*	%	*N*	%
Males *												
WC < 90 (cm)	1317	55.17	1077	53.45	830	47.89	570	51.31	403	52.27	4197	52.35
WC ≧ 90 (cm)	1070	44.83	938	46.55	903	52.11	541	48.69	368	47.73	3820	47.65
Total	2387	100.00	2015	100.00	1733	100.00	1111	100.00	771	100.00	8017	100.00
Females *												
WC < 80 (cm)	1873	35.24	1240	30.91	781	26.95	390	26.02	158	24.23	4441	30.90
WC ≧ 80 (cm)	3440	64.76	2771	69.09	2117	73.05	1109	73.98	494	75.77	9931	69.10
Total	5312	100.00	4011	100.00	2898	100.00	1499	100.00	652	100.00	14372	100.00
Pooled *												
WC < 90/80 (cm)	3189	41.42	2317	38.45	1611	34.79	960	36.78	561	39.42	8638	38.58
WC ≧ 90/80 (cm)	4510	58.58	3709	61.55	3020	65.21	1650	63.22	862	60.58	13751	61.42
Total	7699	100.00	6026	100.00	4631	20.68	2610	100.00	1423	100.00	22389	100.00

Abbreviations: WC, waist circumference. * Prevalence of WC categories were significantly different across all age groups (chi-square test; *p* < 0.05).

**Table 8 healthcare-11-01237-t008:** Categories of WHR in males and females aged 65 years and older (Numbers and percentages).

Variables	Age groups											
	65–69		70–74		75–79		80–84		≧85		Total	
	*n*	%	*n*	%	*n*	%	*n*	%	*n*	%	*N*	%
Males *												
WHR < 0.9	892	37.37	674	33.45	562	32.43	408	36.72	292	37.87	2828	35.28
WHR ≧ 0.9	1495	62.63	1341	66.55	1171	67.57	703	63.28	479	62.13	5189	64.72
Total	2387	100.00	2015	100.00	1733	100.00	1111	100.00	771	100.00	8017	100.00
Females *												
WHR < 0.85	2226	41.91	1494	37.25	893	30.81	408	27.22	129	19.79	5150	35.83
WHR ≧ 0.85	3086	58.09	2517	62.75	2005	69.19	1091	72.78	523	80.21	9222	64.17
Total	5312	100.00	4011	100.00	2898	100.00	1499	100.00	652	100.00	14372	100.00
Pooled *												
WHR < 0.9/0.85	3118	40.50	2168	35.98	1455	31.42	816	31.26	421	29.59	7978	35.63
WHR ≧ 0.9/0.85	4581	59.50	3858	64.02	3176	68.58	1794	68.74	1002	70.41	14411	64.37
Total	7699	100.00	6026	100.00	4631	100.00	2610	100.00	1423	100.00	22389	100.00

Abbreviations: WHR, waist-to-hip ratio. * Prevalence of WHR categories were significantly different across all age groups (chi-square test; *p* < 0.05).

## Data Availability

The data that support the findings of this study are available from the Sports Cloud: Information and Application Research Center of Sports for All, Sport Administration, Ministry of Education in Taiwan but restrictions apply to the availability of these data, which were used under license for the current study, and so are not publicly available. Data are however available from the authors upon reasonable request and with permission of the Sports Cloud: Information and Application Research Center of Sports for All, Sport Administration, Ministry of Education in Taiwan.

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
