# Peer review of "Gender and Age Differences in Anthropometric Characteristics of Taiwanese Older Adults Aged 65 Years and Older"

_healthcare, 2023, doi:10.3390/healthcare11091237_

Round 1
Reviewer 1 Report
1. The amount of data in this study has a certain scale, and the research on anthropometric parameters of the elderly in Taiwan has its research and reference value.
2. In the manuscript, the unit of BMI should be corrected to kg/m2 (2 is superscript).
3. In this large-scale research, the measurement tools used, including the measurement of body circumference, height, and weight, should have uniform specifications of the devices used, and their specifications, brands, and accuracy should be indicated in the manuscript and repeatability.
4. Although body mass index or parameters such as waist circumference, hip circumference, height, and weight have extremely important correlations with body composition, their convenience is very important in epidemiological or large-scale surveys, but if accurate To define nouns, "body composition" in this study should be changed to "anthropometry", which seems more appropriate.
5. There are many data tables in the manuscript, which systematically and clearly present the data of the subjects. However, the large amount of data is not easy for readers to read. It is recommended to streamline or further organize it.
Author Response
Reviewer’s comments to author:
Q1: In the manuscript, the unit of BMI should be corrected to kg/m2 (2 is superscript).
A1: Thanks for your suggestions, and we modified the contexts. Please see lines 130 and 134-135 on page 3, and table 1, 2, 4 and 6.
Q2: In this large-scale research, the measurement tools used, including the measurement of body circumference, height, and weight, should have uniform specifications of the devices used, and their specifications, brands, and accuracy should be indicated in the manuscript and repeatability.
A2: We have reworded the following statement “Body weight and height were recorded in kilograms and meters to the nearest 0.1 kg and 0.1 cm with an electronic height–weight scale, which have passed the official inspection in Taiwan” to indicate the uniform specifications of the devices used. Please see lines 122-124 on page 3.
Q3: Although body mass index or parameters such as waist circumference, hip circumference, height, and weight have extremely important correlations with body composition, their convenience is very important in epidemiological or large-scale surveys, but if accurate To define nouns, "body composition" in this study should be changed to "anthropometry", which seems more appropriate.
A3: Thanks for your suggestions, and we modified the title and contexts of this revised manuscript. Please see this revised manuscript.
Q4: There are many data tables in the manuscript, which systematically and clearly present the data of the subjects. However, the large amount of data is not easy for readers to read. It is recommended to streamline or further organize it.
A4: The personal information in Table 1 overlaps with other tables, so we modified it. Please see this revised manuscript.

Reviewer 2 Report
Generally it's an interesting research idea with an application approach however there are many other effective variables which would be influential in the research objective, so I have had some parts for being added to improve the variability of the content of the research.
1- When you're talking about the age population how have you you classify the age category in your search titleØŸ “Age differences”
2- Check the keywords based on mesh standards.
3- include information of participants in methods of the abstract.
4- Please kindly a state that how how you have done the sampling recruitment in method part of the abstract.
5- The sampling method should be depicted by a Flowchart so that stages of sampling be more clearly understood.
6- The high frequencies of those references, which are related to more than two decades ago, is seen. It's suggested to include the following references as well. Since this factor is highly dependent on exercise behavior, use the following references:
*The effect of aquatic exercise on postural mobility of healthy older adults with endomorphic somatotype.
*Effect of pilates exercises on motor performance and low back pain in elderly women with abdominal obesity.
8- Since the measured variables such as , body mass index (BMI), waist circumference (WC), hip circumference (HC), and waist-to-hip ratio (WHR) would be determinant factors for prediction the most frequently problems in older adults such as fatty liver, blood pressure, so on. Mention it as be done for future research. You can use the following reference:
*Effect of peripheral heart action on body composition and blood pressure in women with high blood pressure." International Journal of Sport Studies for Health 1.2 (2018)..
9- It's recommended that you have some suggestion for future studies regarding your own work for example the Variables that you have measured is highly affected by the aged population lifestyle so you you would focus on some factors such as control and and exercise behavior of aged ones in future studies the following preferences are suggested for that
*The effect of aquatic training on kinematic walking patterns of elderly women.
*Effect of peripheral heart action on body composition and blood pressure in women with high blood pressure." I
11- Please recommend suggestions for further researchers based on the limitations of the research.
12- use more updated references (especially the last three years) for your work
Author Response
Reviewer’s comments to author:
Q1: When you're talking about the age population how have you classify the age category in your search titleØŸ “Age differences”
A1: We classified the 5-year age categories (65–69, 70–74, 75–79, 80–84 and ≥ 85 years) based on Population Data Quarterly of Department of Household Registration, Ministry of the Interior, Taiwan.
Q2 Check the keywords based on mesh standards.
A2: Thanks for your suggestions, and we change the “older adults” to “elderly” based on the mesh standards. Please see the keywords of this revised manuscript.
Q3 include information of participants in methods of the abstract.
A3: We included the demographic information of participants in methods of the abstract. Please see the abstract of this revised manuscript.
Q4 Please kindly a state that how you have done the sampling recruitment in method part of the abstract.
A4: We added the following statement “All participants were recruited using stratified convenience sampling from 46 physical fitness test stations in 22 cities or counties in Taiwan” to indicate the sampling recruitment in method part of the abstract. Please see the abstract of this revised manuscript.
Q5 The sampling method should be depicted by a Flowchart so that stages of sampling be more clearly understood.
A5: There are already 8 tables in this paper, so we decided to state the the sampling process in words and cite the previously published literature in order to avoid too many tables/charts.
Q6 The high frequencies of those references, which are related to more than two decades ago, is seen. It's suggested to include the following references as well. Since this factor is highly dependent on exercise behavior, use the following references:
*The effect of aquatic exercise on postural mobility of healthy older adults with endomorphic somatotype.
*Effect of pilates exercises on motor performance and low back pain in elderly women with abdominal obesity.
A6: Thanks for your suggestions, and we updated some of our references and included your recommendation of references.
Q7 Since the measured variables such as, body mass index (BMI), waist circumference (WC), hip circumference (HC), and waist-to-hip ratio (WHR) would be determinant factors for prediction the most frequently problems in older adults such as fatty liver, blood pressure, so on. Mention it as be done for future research. You can use the following reference:
*Effect of peripheral heart action on body composition and blood pressure in women with high blood pressure." International Journal of Sport Studies for Health 1.2 (2018).
A7: Thanks for your suggestions, and we modified the context and included your recommendation of references. Please see lines 343-355 on page 14.
Q8 It's recommended that you have some suggestion for future studies regarding your own work for example the Variables that you have measured is highly affected by the aged population lifestyle so you would focus on some factors such as control and exercise behavior of aged ones in future studies the following preferences are suggested for that
*The effect of aquatic training on kinematic walking patterns of elderly women.
*Effect of peripheral heart action on body composition and blood pressure in women with high blood pressure."
A8: Thanks for your suggestions, and we added some suggestion for future studies and included your recommendation of references. Please see this revised manuscript.
Q9: Please recommend suggestions for further researchers based on the limitations of the research.
A9: Thanks for your suggestions, and we added one paragraph and included your recommendation of references. Please see lines 343-353 on page 14.
Q10: use more updated references (especially the last three years) for your work
A10: Thanks for your suggestions, and we updated and added some references which are mostly updated. Please see this revised manuscript.

Reviewer 3 Report
Thank you for the opportunity to review of the article titled "Anthropometric characteristics and body composition of Taiwanese older adults: Age and gender differences". This study aimed to describe gender- and age-specific distributions of anthropometric measurements and body composition of Taiwanese older adults. This study has several concerns need to be addressed.
Page 13, lines 3-5, and the abstract: The authors state that “The results showed statistically significant differences in weight, height, WC, HC, and WHR among Taiwanese individuals aged 65 years and older”. Does this mean that there were significant differences between age groups?
Page 13, lines 5-6, and the abstract: The authors state that “there was no significant difference in BMI between men and women”; however, Table 1 shows a difference between males and females. Is this an error?
Page 13, lines 21-22: The authors describe that “A previous study showed that women presented a higher prevalence of obesity among Mexican older adults [21]; the same results were found in American adults [22, 23]. However our results show that the percentages of males and females in these two categories are not very different. We conclude that this may be related to the imbalance of the male to female ratio in our sample (approximately 1: 2 male to female ratio)”. The authors have calculated the percentage of overweight and obesity in males and females respectively, so the imbalance in the sex ratio could not be the reason for this. I would like to see a detailed discussion of sex differences and comparisons between other countries and Taiwan.
Page 13, lines 40-41: The authors describe that “Overall, our data provide WC, HC, and WHR data as a reference. This can reveal possible health risks to Taiwanese older adults”. However, health risks cannot be derived from the results of this study. As the authors also state in the limitations of the study, this is a cross-sectional study that does not even include variables related to disease or health risks. Please discuss by citing previous studies that support the identification of health risks.
Overall: Sex expressions were different, i.e., men and male, women and female; please standardize.
Author Response
Reviewer’s comments to author:
Q1: Page 13, lines 3-5, and the abstract: The authors state that “The results showed statistically significant differences in weight, height, WC, HC, and WHR among Taiwanese individuals aged 65 years and older”. Does this mean that there were significant differences between age groups?
A1: We reworded the statement to be “The results showed statistically significant differences in weight, height, WC, and WHR across all age groups among both male and female individuals aged 65 years and older in Taiwan.”. Please see the abstract of this revised manuscript and lines 272-274 on page 13.
Q2: Page 13, lines 5-6, and the abstract: The authors state that “there was no significant difference in BMI between men and women”; however, Table 1 shows a difference between males and females. Is this an error?
A2: We reworded the statement to be “However, there was no significant difference in BMI and HC between males and females in all age groups.”. Please see the abstract of this revised manuscript and lines 274-275 on page 13.
Q3: Page 13, lines 21-22: The authors describe that “A previous study showed that women presented a higher prevalence of obesity among Mexican older adults [21]; the same results were found in American adults [22, 23]. However, our results show that the percentages of males and females in these two categories are not very different. We conclude that this may be related to the imbalance of the male to female ratio in our sample (approximately 1: 2 male to female ratio)”. The authors have calculated the percentage of overweight and obesity in males and females respectively, so the imbalance in the sex ratio could not be the reason for this. I would like to see a detailed discussion of sex differences and comparisons between other countries and Taiwan.
A3: Thank you for pointing out this issue. We deleted the conclusion sentence and cited some previous studies to discuss sex differences and comparisons between other countries and Taiwan. Please see this revised manuscript on page 13 lines 289-292.
Q4: Page 13, lines 40-41: The authors describe that “Overall, our data provide WC, HC, and WHR data as a reference. This can reveal possible health risks to Taiwanese older adults”. However, health risks cannot be derived from the results of this study. As the authors also state in the limitations of the study, this is a cross-sectional study that does not even include variables related to disease or health risks. Please discuss by citing previous studies that support the identification of health risks.
A4: Thank you for this suggestion. We cited some previous studies and made some discussions. Please see this revised manuscript on page 13 lines 311-316.
Q5 Overall: Sex expressions were different, i.e., men and male, women and female; please standardize.
A5: We have made some changes in this revised manuscript as your kind suggestion. Please see this revised manuscript.
